# Clinicopathological and Molecular Features of Secondary Cancer (Metastasis) to the Thyroid and Advances in Management

**DOI:** 10.3390/ijms23063242

**Published:** 2022-03-17

**Authors:** Marie Nguyen, George He, Alfred King-Yin Lam

**Affiliations:** 1Cancer Molecular Pathology, School of Medicine and Dentistry, Menzies Health Institute Queensland, Griffith University, Gold Coast 4222, Australia; nguyenmari20@gmail.com (M.N.); hegeo924@gmail.com (G.H.); 2Pathology Queensland, Gold Coast University Hospital, Southport 4215, Australia

**Keywords:** thyroid, metastases, secondary, pathology, prognosis, treatment

## Abstract

Secondary tumours to the thyroid gland are uncommon and often incidentally discovered on imaging. Symptomatic patients often present with a neck mass. Collision tumours of secondary tumours and primary thyroid neoplasms do occur. Ultrasound-guided fine-needle aspiration, core-needle biopsy, and surgical resection with histological and immunohistochemical analysis are employed to confirm diagnosis as well as for applying molecular studies to identify candidates for targeted therapy. Biopsy at the metastatic site can identify mutations (such as EGFR, K-Ras, VHL) and translocations (such as EML4-ALK fusion) important in planning target therapies. Patients with advanced-stage primary cancers, widespread dissemination, or unknown primary origin often have a poor prognosis. Those with isolated metastasis to the thyroid have better survival outcomes and are more likely to undergo thyroid resection. Systemic therapies, such as chemotherapy and hormonal therapy, are often used as adjuvant treatment post-operatively or in patients with disseminated disease. New targeted therapies, such as tyrosine kinase inhibitors and immune checkpoint inhibitors, have shown success in reported cases. A tailored treatment plan based on primary tumour features, overall cancer burden, and co-morbidities is imperative. To conclude, secondary cancer to the thyroid is uncommon, and awareness of the updates on diagnosis and management is needed.

## 1. Introduction

Secondary tumours of the thyroid gland are uncommon and seldom presents symptomatically [1]. Many cases originate from the kidney, lung, and breast [2]. Direct or metastatic spread from the upper aerodigestive tract and head/neck region is also common [3]. A 2017 meta-analysis by Straccia and colleagues examines primary sites, diagnosis, and management [2]. In recent years, however, there have been technological advances in the diagnosis and management of thyroid tumours. These include the use of computer-aided diagnosis systems, molecular and genetic profiling, and targeted therapy, all of which have not been adequately reflected in present-day literature. Complexities in the diagnostic workup of secondary thyroid cancers, including common discrepancies on histopathology and the presence of collision tumours, will be explored. Metastasis from distant sites will be the focus of this review, and haematological primary cancers such as lymphomas were not included. This review aims to provide clinicians with a comprehensive update regarding the clinicopathological features, diagnostic investigations, and current management of secondary cancer to the thyroid gland.

## 2. Clinical Features

Secondary thyroid cancer may be suspected in the setting of a known history of malignancy. Choi and colleagues in 2016 found that 78.3% of patients had a known primary tumour at another site [4]. A thorough clinical history would involve exploring any previous surgery or radiation, relevant investigations, and a systemic review of these features [5]. Systemic symptoms such as weight loss, fatigue, and night sweats would raise the suspicion of an underlying malignancy [6].

The clinical presentation of secondary thyroid cancer varies widely depending on the underlying primary cancer. Metastases to the thyroid gland often represent one of several metastatic sites [6,7,8]. As such, features of other sites of metastases should be noted and investigated accordingly. Examples include dyspnoea secondary to pleural effusion from squamous cell carcinoma of the lung, or abdominal pain, weight loss, and jaundice from a pancreatic primary carcinoma [9].

The thyroid is the sole metastatic site in up to 72% of cases [10]. Table 1 depicts characteristics of patients with isolated metastasis to the thyroid gland from recent clinical series. Literature is limited as many studies do not include figures regarding sole metastases to the thyroid. From the small number of clinical series, there is no obvious trend in primary sites for patients with isolated metastasis to the thyroid. Nevertheless, many of the solitary metastases are from the kidney or lung, representing the prevalence of common primary cancers metastasising to the thyroid gland. Despite heterogeneous data regarding survival amongst various studies, it appears that patients with isolated metastasis generally have improved survival compared to those with metastases elsewhere (widely disseminated disease). These values are of clinical significance as patients with isolated metastases are more likely to be surgical candidates for thyroid resection [11]. The six patients with solitary tumour metastases from 30 cases (20%) of thyroid metastases reported by Ghossein and colleagues all underwent thyroid resection and achieved excellent survival rates [12]. In comparison, all 13 cases of secondary thyroid cancer reported by Kim et al. had widespread dissemination of disease on presentation. None of these patients received surgery, and the median survival was 6 months (range 1 to 16 months) [13]. As isolated metastasis to the thyroid is a better prognosis, detection of this requires urgent management before further dissemination occurs.

Isolated metastases may be the initial manifestation of an underlying primary malignancy [14]. In a study by Rahman and colleagues, 2 out of 5 patients presented with thyroid metastases prior to the discovery of a renal and pancreatic primary [9]. Similarly, the primary cancers of 2 patients out of 91 (melanoma and squamous cell carcinoma of the pharynx) were identified only after the initial presentation of a thyroid metastasis [11]. Wood and colleagues revealed 5 initial thyroid presentations out of 15 cases, including primaries from renal cell carcinoma, lung large cell carcinoma, retroperitoneal liposarcoma, Merkel cell carcinoma of the forearm, and adenocarcinoma of unknown origin [14].

Many cases of secondary thyroid cancer are discovered incidentally during follow-up of the primary malignancy or an unrelated condition. Approximately 28% (10/36) and 31% (28/90) of cases reported by Papi et al. in 2007 and Romero Arenas et al. in 2014, respectively, were found as incidental thyroid masses on further investigations [6,7]. Similarly, 54% (15/25) of patients were incidentally discovered during the follow up of primary cancer, as reported by Calzolari and colleagues in 2008 [15]. Wood et al. reported an incidental case found during a neck dissection for lymphadenopathy [14].

Kim and colleagues illustrated differing clinical presentations of secondary thyroid cancer amongst 22 patients: 14 had palpable nodules on clinical examination while 8 were incidentally discovered during the workup of a primary tumour (ultrasound [US], FDG-PET [fluorodeoxyglucose-positron emission tomography], CT [computed tomography]) [16]. As small lesions are often asymptomatic, a high index of suspicion is recommended for patients with a previous history of malignancy [17]. The high proportion of secondary thyroid cancer diagnosed in asymptomatic patients in the recent series reflects the advancement of various diagnostic techniques (i.e., ultrasound, CT, FDG-PET, FNA) [18,19].

**Table 1 ijms-23-03242-t001:** Characteristics of patients with sole/isolated metastasis to the thyroid gland, as described in recent clinical series published between 2005 to 2021.

Author (Year)	No. of Cases of Isolated Metastasis/Total STC Cases (%)	Primary Sites	Treatment	Survival for Isolated Metastasis to Thyroid	Survival for Widely Disseminated Disease
Stergianos et al. (2021)	5/31 (16.13%)	3 RCC, 1 melanoma, 1 SCC of unknown primary origin	Thyroid resection: in 4 isolated metastasis cases (all except SCC of unknown origin)	Not specified by authors	Not specified by authors [20]
Ghossein et al. (2020)	6/30 (20%)	4 RCC, 1 oesophageal adenocarcinoma, 1 retroperitoneum leiomyosarcoma	Thyroid resection: all 6 isolated metastasis cases	3-, 5-, and 10-year specific survival was 100% for all	3-, 5-, and 10- year disease-specific survival was 56%, 41%, and 14%, respectively [12]
Wang et al. (2018)	2/21 (9.52%)	Not specified by authors	Not specified by authors	Not specified by authors	Not specified by authors [21]
Zhang et al. (2017)	17/32 (53.13%)	Not specified by authors	Thyroid resection: 8 isolated metastasis cases No surgery: 9 isolated metastasis cases (other treatments not specified by authors)	7 cases were alive at time of follow up (time frame not provided), 3 of which had surgery and 4 had no surgery	3 cases were alive at time of follow up (time frame not provided) [22]
Romero Arenas et al. (2014)	11/90 (12.22%)	7 lung carcinoma, 3 RCC, 1 not specified by authors	Not specified by authors	Not specified by authors	Not specified by authors [6]
Kim et al. (2014)	0/13 (0%)	N/A	No surgery: all 13 cases	N/A	Median survival 6 months (range 1–16) [13]
Ishikawa et al. (2011)	0/4 (0%)	N/A	Thyroid resection: all 4 cases Adjuvant chemotherapy in 1 case	N/A	Median survival after thyroidectomy was 10 months for 3 cases (range 3–23); one case receiving adjuvant chemotherapy was disease-free at 1 year [8]
Papi et al. (2007)	5/36 (13.89%)	2 RCC, 1 urothelial sarcomatoid carcinoma, 2 not specified by authors	Not specified by authors	3 patients alive at end of study (time frame not specified)	1 patient alive at end of study (time frame not specified) [7]
Mirallié et al. (2005)	21/29 (72.41%)	Not specified by authors	Not specified by authors	11 deceased (mean delay 2.1 years), 7 alive without disease, 3 were alive with disease	7 deceased (mean delay 6 months), 1 survived with short follow-up due to progressive tumour [10]
Kim et al. (2005)	7/22 (31.82%)	2 oesophageal SCC, 1 breast ductal carcinoma, 1 lung adenocarcinoma, 1 uterine cervix SCC, 1 pulmonary artery intimal sarcoma, 1 biliary tract adenocarcinoma	Thyroid resection: 1 isolated metastasis case Chemotherapy: 1 isolated metastasis case No treatment: 5 isolated metastasis cases	3 patients alive at 4/3/2 months (includes a thyroidectomy and a chemotherapy patient), 3 deceased at 4/6/16 months, 1 deceased in less than 1 month	4 alive at average 12 months (range 6–17)—3 received chemotherapy, and 1 underwent surgery; 11 deceased with a median survival of 8 months (range < 1–34)—5 received no treatment, 3 chemotherapy, 1 interferon-α, 1 surgery and interferon-α [16]

STC, secondary thyroid cancer; RCC, renal cell carcinoma; SCC, squamous cell carcinoma.

The symptoms of secondary thyroid cancer are variable and largely resemble that of primary thyroid cancers on clinical presentation. A neck mass is the most common reported symptom and may be characterised as a single nodular tumour, multiple nodules, or a goitre [6]. Hegerova and colleagues noted that 95% (92/97) of symptomatic cases presented with either a palpable nodule, firm thyroid mass, or goitre [11]. Three cases identified by Vardar and colleagues presented with a two-month history of a growing thyroid mass [23]. Similarly, Papi et al. observed 72% of patients (26/36) presenting with rapidly enlarging nodules. Amongst these patients, 56% had a single nodule, while 44% had multiple nodules [7]. A recent literature review observed that metastases to the thyroid from solid malignancies were more likely to produce a nodular infiltration pattern compared to haematological malignancies, which infiltrated diffusely, resulting in goitre [24]. As there are many similarities between the presentation of primary and secondary thyroid cancer, a thorough clinical history, including previous malignancy, is of utmost importance.

Lymphadenopathy in surrounding areas (i.e., cervical or subclavian) is well documented in studies. Approximately 66% (24/36) of patients had lymph node involvement, as described by Papi et al. [7]. Kim et al. found cervical lymphadenopathy present in 92% (12/13) of secondary thyroid cancer cases with diffuse thyroid involvement [13]. All patients in the study had extensive metastases to other organs at the time of diagnosing secondary thyroid cancer. When comparing thyroid tumour type, Saito et al. found lymphadenopathy in 100% of diffuse cases in contrast to 33% amongst those with nodular tumours [19].

Symptoms related to mass effects, such as cough, dysphagia, dyspnoea, neck pain, and dysphonia, are infrequent yet important to recognise as they may require immediate management. Khaddour et al. in 2019 found that masses larger than 4 cm were more likely to present symptomatically. These appear to be more common in thyroid metastases at an advanced stage [25]. Zhang et al. in 2017 reported that 21.9% of patients (7/32) presented with either an enlarged gland, dysphagia, dysphonia, or a combination of these. Out of 90 cases described by Romero Arenas et al. in 2014, 9% had dysphagia and 9% had a cough. A systematic review analysing 147 patients with thyroid metastasis from clear cell renal cell carcinoma reported dysphagia in 8.8%, dyspnoea in 6.1%, hoarseness in 4%, neck pain in 3.4% and cough and stridor in 2.7% [17]. Similarly, Hegerova and colleagues reported dyspnoea or tracheal compression in 3.1% of cases (3/97) [11]. In one study, four out of ten cases of secondary thyroid cancer presented with unilateral vocal cord palsy secondary to metastatic recurrent laryngeal nerve infiltration [26]. As obstructive symptoms can mimic the presentation of primary anaplastic thyroid cancer, it is necessary to distinguish this from potential secondary thyroid cancer. Other rare symptoms include haemoptysis, epistaxis, acute respiratory failure, and chest pain [15,17].

Most patients with secondary metastasis are euthyroid; however, abnormal thyroid function and coexistent thyroid pathology have been described in several cases [18,27]. It is believed that the mechanism of hyperthyroidism is the parenchymal destruction and consequent leakage of the stored hormones into the circulation secondary to metastatic infiltration or embolization [7]. Hypothyroidism may occur if thyroid function does not recover [24]. In a study by Kim and colleagues, three out of twelve cases were found to have hypothyroidism. In all three cases, metastases involved both thyroid lobes [13]. A recent literature review identified 26 cases of hyperthyroidism associated with secondary thyroid metastases [24]. Amongst these were metastases from breast cancer (*n* = 6), lung cancer (*n* = 5), and various haematological malignancies (*n* = 7) [24]. The degree of hyperthyroidism ranged from subclinical to severe, with features of thyroiditis documented in some patients (i.e., painful goitre). Overall, thyrotoxicosis is rare, while hypothyroidism secondary to massive infiltration may take months to develop [13,27]. Hence, it is recommended that all patients with suspected secondary metastases undergo thyroid function tests.

Many thyroid pathologies, including secondary thyroid cancer, may present as nodular or diffuse involvement of the thyroid [19] and should be included in the differential diagnosis. Solitary thyroid nodules can be categorised as benign or malignant based on clinical presentation, radiological features, and cytological assessment. Benign nodules include adenomas, hyperplasia, cysts, colloid nodules, infectious nodules, lymphocytic or granulomatous nodules, and congenital abnormalities. Differentials for diffuse thyroid disease include malignancy, autoimmune thyroiditis (Hashimoto thyroiditis), subacute thyroiditis (de Quervain), acute suppurative thyroiditis, Graves’ disease, and multinodular goitre [13].

High clinical suspicion in diagnosing secondary thyroid cancer is required as metastases are most often misinterpreted as primary thyroid cancers [3,11,27]. Eliciting a history of nonthyroidal malignancy may provide a clue to distant origin. Overall, diagnosing secondary thyroid cancer requires correlation between clinical, radiological, and histological findings with the aid of immunophenotypic and molecular profiles in diagnostically challenging cases.

## 3. Imaging

The diagnostic workup of secondary thyroid cancer is like that of a common thyroid nodule. The combination of clinical history and histology often leads to an accurate diagnosis. Although imaging can assist with the diagnostic process, results are usually non-specific and cannot differentiate primary from secondary cancer. Early detection of secondary thyroid cancer is important, especially in isolated metastases, as it has a direct impact on prognosis and management.

### 3.1. Ultrasound

Ultrasonography is often the first line initial tool used to investigate thyroid disease. It is convenient, non-invasive, and relatively cost-effective. Saito et al. reported ultrasound to be diagnostic for malignancy in 96% of cases in 2014. Although helpful in assessing suspicious lesions, sonographic features alone cannot differentiate between primary thyroid cancer and metastatic cancer [28]. In 2018, Falcone and colleagues examined four different ultrasound risk-assessment systems for secondary thyroid malignancy (American Association of Clinical Endocrinologists (AACE), American Thyroid Association (ATA), European Thyroid Imaging Reporting and Data System (EU-TIRADS), Korean Thyroid Imaging Reporting and Data System (K-TIRADS) [28]. The authors concluded that all four systems were able to consistently categorise metastatic lesions in the thyroid gland as suspicious enough to warrant fine needle aspiration (FNA) biopsy. The American College of Radiology Thyroid Imaging Reporting and Data System (ACR TIRADS) was not examined in this study but has been reported to have the greatest overall performance compared to other classification systems, resulting in lower rates of false positives and less unnecessary FNA biopsies [29]. With the increased use of ultrasonography, the challenge is to avoid overdiagnosis and overtreatment. Hence, FNA biopsy should be restricted to lesions that are suspicious for malignancy based on clinical history, physical examination, or imaging.

Most secondary thyroid metastases are described to be solid, hypoechoic lesions with ill-defined margins [4,7,19,28]. Such lesions can be diffuse or nodular, and nodules may be solitary or multiple in number [19,22,27]. In 2014, Kim et al. noted a reticular pattern of internal hypoechoic lines without increased vascularity to be a unique feature of diffuse metastasis to the thyroid. These lines are believed to represent intra-thyroidal lymphatics distended with malignant cells [13]. Hyperechoic areas within hypoechoic regions usually represent calcified or necrotic lesions [19]. Punctate calcifications are more commonly observed in primary thyroid carcinomas [7]. However, 87% of cases reported by Choi et al. did not show calcifications on ultrasound [4]. Intranodular vascularity is frequently reported on Doppler ultrasonography [7,17,19,22]. One large study reported a significant association between intra-nodular vascularity and malignant nodules; however, this could not distinguish primary from secondary thyroid cancer [30]. Abnormally enlarged lymph nodes were visualised in roughly half of all cases [4,28]. A systematic review focusing on metastasis of renal cell carcinoma to the thyroid reported that 4.8% of patients had concomitant internal jugular thrombosis on ultrasound [17]. This was likely secondary to either hypercoagulable state or local tumour invasion.

Elastography is a sonographic method of estimating tissue hardness that has been proposed to assist with differentiating benign from malignant thyroid lesions and evaluating optimal sites for biopsy [31,32]. In strain elastography, the stiffness of the lesion is compared to normal thyroid tissue through external compression. This technique is operator dependent and cannot be used in thyroids with diffuse disease. Conversely, shear-wave elastography (SWE) estimates stiffness by measuring the velocity of wave propagation. However, there appears to be no statistically significant difference in diagnostic performance when using ultrasound combined with SWE compared to ultrasound alone to distinguish benign from malignant lesions [32]. Elastography is further limited by poorly defined cut-off values for interpreting stiffness. There have been very few reports on the use of elastography in the evaluation of secondary thyroid metastases and results are mixed [31,33].

### 3.2. Other Imaging Modalities

The diagnostic performance of ultrasound depends on operator skill and experience. CT, magnetic resonance imaging (MRI), and PET-CT are examples of other imaging modalities that can be utilised to detect or further evaluate metastases. Out of 11 cases of metastatic cancer to the thyroid, a thyroid malignancy was initially detected with ultrasound in three patients, CT in four, and four through clinical history and examination [34]. On CT, thyroid metastases are often reported as heterogeneous and hypodense relative to the surrounding normal thyroid tissue [2]. Battistella et al. described the use of PET-CT in 6 patients with suspicious thyroid FNA results to exclude other metastatic sites. In these six patients, PET-CT confirmed a focal area of increased activity in the thyroid [27]. Conversely, Agrawal et al. presented a case in which secondary thyroid metastasis was discovered on 18F-FDG PET-CT during the evaluation of recurrence of lung non-small cell carcinoma [35]. 68Ga-DOTATATE PET/CT has also been shown to detect secondary thyroid cancer outside of its common use in the investigation of neuroendocrine tumours [36]. Metastatic lesions, however, can appear benign on PET-CT due to non-specific metabolic activity [17]. The size of such metastatic lesions in the thyroid range between 1 to 7 cm, with an average size of 3 to 4 cm on imaging [3,11,37].

Computer-aided diagnosis (CAD) systems have progressed over the years with the aim of improving diagnostic performance through overcoming the heterogeneity and subjectivity of human radiologists. Classic machine learning is based on explicit features of an image labelled by human experts, whereas deep neural networks (deep learning) can automatically extract more implicit features from an image by learning complex patterns in large datasets. A systematic review found the diagnostic performance of CAD systems for malignant thyroid nodules on static ultrasound images like that of experienced radiologists [38]. However, more recent studies have shown improved diagnostic performance with ultrasound CAD systems [39,40]. Radiologists outperformed CAD systems during real-time clinical diagnosis [38]. As CAD systems advance, processing capabilities can extend to the level of identifying the likelihood of disease, grading disease, or even providing recommendations. Hence, CAD systems have a potential role in assisting clinicians in diagnosis and decision-making. However, as there are currently no studies on the use of CAD systems to differentiate between primary thyroid cancer and metastases to the thyroid, this represents a potential area for future research.

## 4. Biopsy

Ultrasound-guided FNA or core needle biopsy (CNB) are being used to diagnose secondary thyroid lesions. Sometimes, surgical specimen resection is needed to confirm the diagnosis. Out of these three methods, FNA is the standard diagnostic modality for thyroid nodules. Within the studies published between 1995 and 2015, FNA was the most frequent tool used to diagnose the original primary site of secondary thyroid cancer (89%), followed by surgical specimen histology (11%) [2]. In FNA, secondary thyroid cancer often shows abundant cellularity, and cells may resemble those of the original primary site.

The proportion of correct cytological diagnoses is heterogeneous and dependent upon various factors. In the settings of known primary malignancies, FNA is useful in making the diagnosis of metastatic cancer in the thyroid gland. FNA has been shown to be successful in correctly identifying secondary thyroid cancer in 46% to 94% of cases (see Table 2). Hegerova and colleagues concluded that FNA was sensitive (94%) and specific (100%) for detecting metastases [11]. On the other hand, Choi et al. reported low sensitivity (58.6%) yet high specificity (100%) for detecting secondary thyroid cancer [4]. The accuracy of FNA results may relate to the metastatic tumour origin. Kim et al. reported high accuracy with FNA in diagnosing thyroid metastasis from lung cancer (90.1%) [13]. In contrast, 28.7% of patients with metastasis from renal cell carcinoma were incorrectly diagnosed [13]. Similarly, FNA was diagnostic for renal cell carcinoma metastasis to the thyroid in only 29.4% of patients, according to a systematic review by Khaddour and colleagues in 2019 [17]. Interestingly, metastases originating from the breast, lung, and colon are associated with a higher percentage of correct cytological diagnoses when compared to those originating from the oesophagus, cervix, and kidney [41].

As shown in Table 2, inaccurate FNA results for secondary thyroid cancer were frequently interpreted as either non-diagnostic, benign, or primary thyroid cancer. The non-diagnostic rate of FNA for secondary thyroid cancer ranges between 8.0% and 9.8% due to scant or absent cytological material [3,4]. This is lower than the non-diagnostic rate for thyroid FNA in general and requires repeat diagnostic examination to evaluate for possible malignancy [42,43]. Conversely, the non-diagnostic rate for a primary renal cell carcinoma was higher at 47.1% [17]. This likely relates to its unique microscopic features.

As most guidelines recommend FNA as the first-line diagnostic investigation for thyroid lesions, the indication for core-needle biopsy (CNB) is not well established [44]. CNB appeared to outperform FNA with a sensitivity of 100% compared to 58.6% (*p* = 0.008), indicating the value of CNB in the workup of thyroid lesions suspicious for metastasis [4]. The higher sensitivity of biopsy, when compared to cytology, is related to the fact that biopsy provides information on tissue architecture and material for ancillary studies.

Recommendations for CNB as an alternative first-line diagnostic modality stems from the reduction in non-diagnostic results and unnecessary diagnostic thyroidectomies [45]. Studies have suggested the use of CNB in cases of non-diagnostic FNA results and follicular lesions or atypia of undetermined significance [44]. Despite such findings, FNA continues to be the first line as it is simple, safe with rare complications, and cost-effective whilst maintaining high specificity. On the other hand, known complications following CNB include pain, bleeding, infection, haematoma, and injury to nearby structures. Although complication rates are relatively low (0.5–1%), other considerations include technical feasibility considered (i.e., lesion not within reach or located near vital structures) and the need for an experienced operator [46]. Care should be taken when performing CNB of the thyroid due to the presence of several vascular structures and nerves adjacent to the thyroid gland. Lesions should always be sampled under imaging guidance to avoid inadvertent injury and error in sampling [34].

Rapid on-site evaluation (ROSE) of fine needle aspiration of the thyroid gland is available in some centres where pathology staff is available on site. The procedure involves a staff member (either cytotechnician or pathologist) doing a quick staining of the sample(s) at the time of fine-needle aspiration. This allows sample adequacy to be checked so that additional samples can be taken if the first sample is not adequate. It also provides a preliminary diagnosis so that further specimens can be taken for cell block preparation and ancillary studies [47]. There is reasonable evidence that ROSE improves the adequacy and diagnostic yield of thyroid biopsies; hence, some authors have recommended its use with FNA and CNB [9]. However, as ROSE is a labour- and time-intensive procedure and associated with increased operational costs, it is not widely utilised in clinical practice outside tertiary institution service centres.

## 5. Macroscopic Features

Metastasis may appear diffuse, nodular, or continuous with adjacent organs and may affect the thyroid gland bilaterally or unilaterally [18]. Nodular goitres may be difficult to distinguish from multinodular metastasis [6]. Conversely, direct extension from adjacent organs is relatively simple to identify. Secondary thyroid malignancies may appear macroscopically like the corresponding primary tumour. For example, renal cell carcinoma metastasis has been described as a yellow and haemorrhagic mass on the thyroid, reflecting its usual appearance at its primary site [48]. A gastrointestinal primary adenocarcinoma may present with mucin pools in metastatic foci [48].

## 6. Collision Tumours

Collision tumours, also known as tumour-to-tumour metastases, occur when a primary malignant tumour (donor) metastasises to another tumour (recipient) in an organ. The recipient may be benign or malignant. As a rare phenomenon, collision tumours are comprised of at least two discrete cell populations that have distinct borders and should be considered when dimorphic tumour patterns are encountered [49,50]. Examples listed in Table 3 demonstrate the recipient as a thyroid neoplasm and the donor as true metastasis without direct extension.

Collision of metastatic cancer to the thyroid gland could occur with benign or malignant thyroid lesions. Table 3 illustrates characteristics of tumour-to-tumour metastases, as described in various case reports and case series. Follicular adenoma is the most common recipient (17 cases), followed by papillary thyroid carcinoma and follicular variant of papillary carcinoma (7 cases each). Overall, low-risk recipient tumours (*n* = 29) outnumber high-risk recipients (*n* = 10). Recent literature has mirrored these findings, noting that collision tumours often occur in benign or low-risk follicular cell-derived thyroid neoplasms [5]. These neoplasms include follicular adenoma, oncocytic adenoma, non-invasive follicular thyroid neoplasm with papillary like nuclear features (NIFTP), and follicular variants of papillary thyroid carcinoma (many cases would also be classified as NIFTP by revised WHO classification). Rarely, collision tumours occur with malignant thyroid neoplasms such as papillary thyroid carcinoma, follicular thyroid carcinoma, and oncocytic thyroid carcinoma. Only one case of follicular carcinoma in the literature has been reported as a recipient [51]. Herein, there is another case showing a collision tumour of follicular carcinoma and metastatic renal cell carcinoma highlighted by immunohistochemical staining (Figure 1).

No high grade (poorly differentiated thyroid carcinoma) or anaplastic follicular cell-derived thyroid carcinoma (anaplastic thyroid carcinoma or squamous cell carcinoma) were noted as collision tumours. The high number of benign recipients may relate to its slow disease progression, which allows time for extrathyroidal metastases to reach and develop in the thyroid gland. The unusual presentation of collision tumours is a diagnostic challenge, resulting in possible misdiagnosis and delayed treatment. Immunohistochemistry can be useful in differentiating between different neoplasms in collision tumours [52]. In addition, for both benign and malignant thyroid lesions, the most common primary site for secondary thyroid cancer is either the lung or kidney, reflecting the higher prevalence of this cancer to metastasise to the thyroid gland.

**Table 3 ijms-23-03242-t003:** Collision tumours of secondary cancer in the thyroid and primary thyroid neoplasm described in literature.

Thyroid Tumour	Primary Tumour with Number of Cases	Total No. of Cases
Follicular adenoma	4× Lung adenocarcinoma [3,53,54,55] 2× Lung small cell carcinoma [56,57] 1× Poorly differentiated lung carcinoma with basaloid pattern [50] 3× Renal cell carcinoma [58,59,60] 2× Breast carcinoma [61,62] 2× Colon adenocarcinoma [55,63] 1× Prostate adenocarcinoma [61] 1× Oesophageal adenocarcinoma [12] 1× Melanoma [3]	17
Oncocytic (Hürthle cell) adenoma	1× Breast phyllodes tumour [64] 1× Renal cell carcinoma [65] 1× Endometroid adenocarcinoma [66] 1× Gastric adenocarcinoma [67]	4
Non-invasive follicular thyroid neoplasm with papillary like nuclear features (NIFTP)	1× Renal cell carcinoma [12]	1
Follicular variant of papillary carcinoma	3× Renal cell carcinoma [52,68,69] 1× Lung adenocarcinoma [70,71] 1× Lung small cell carcinoma [69] 1× Breast carcinoma [68]	7
Papillary carcinoma	1× Colorectal adenocarcinoma (site not specified) [12] 1× Rectal adenocarcinoma [72] 2× Renal cell carcinoma [73,74] 1× Breast carcinoma [5] 1× Lung adenocarcinoma [75] 1× Melanoma [20]	7
Follicular carcinoma	Cutaneous malignant melanoma [51]	1
Oncocytic (Hürthle cell) carcinoma	1× Colon adenocarcinoma [76] 1× Renal cell carcinoma [77]	2

## 7. Microscopic Features

Secondary thyroid malignancies can often be distinguished from primary thyroid cancers through cytoplasmic, nuclear, and background features [3]. Atypical features that do not align with characteristics of primary thyroid neoplasms should raise the suspicion of metastasis. Diffuse secondary thyroid cancer is likely to only show malignant cells, while focal metastases may show both malignant and normal follicular cells on cytology [47]. As primary thyroid neoplasms are often low grade, metastasis should be considered in the presence of significant atypia or a distinct population of malignant cells bearing little resemblance to follicular epithelium. In general, high cellularity, focal tumour necrosis, and high-grade nuclear features are frequent findings in secondary thyroid cancers [2]. However, Pusztaszeri et al. noted high variability in cellularity and the sizes of cell clusters. Along with a thorough clinical history, pathologists should be familiar with cytological features of common primary cancers related to secondary thyroid cancer.

Diagnostic problems are encountered when there are overlapping similarities between secondary thyroid metastases and primary thyroid malignancies. Metastases can mimic alveolar or microfollicular structures seen in thyroid tumours. In several cases, FNA was unable to differentiate between metastatic malignancy in the thyroid gland and primary anaplastic thyroid carcinoma [2,4,11]. Only 20% to 30% of anaplastic follicular cell-derived thyroid carcinomas stain for thyroglobulin [7]. Clear cells that are present in renal cell carcinomas and some breast carcinomas may also be observed in primary thyroid neoplasms. These clear cells can be arranged in alveolar or glandular structures resembling follicles [2]. Cytological features of renal cell carcinoma and Hürthle (oncocytic) tumour cells have been reported to bear similarities [3]. Correspondingly, 3 of 9 renal cell carcinoma metastases of the thyroid gland, reviewed by Pusztaszeri et al., were misinterpreted as follicular thyroid neoplasms on FNA [3]. Renal cell carcinoma primaries appear to be most frequently associated with inaccurate FNA results in the literature [47]. Adenoid cystic carcinoma is a malignant salivary gland-type tumour that often originates from the head and neck region. It has been misclassified as papillary thyroid carcinoma and poorly differentiated thyroid carcinoma (follicular-derived carcinoma, high grade) due to its rarity and the presence of basaloid and microfollicular features [3]. Cytological diagnosis can also be limited by haemorrhage and necrosis, leading to possible misdiagnosis or non-diagnostic results. Hence, clinical history and ancillary studies are important in overcoming such diagnostic issues.

## 8. Pathological Investigations

Ancillary studies play an important role in metastases of unknown origin as the early and accurate identification of malignancy dictates clinical decision making and treatment options. It may be used to diagnose challenging lesions and guide treatment. Immunohistochemistry has been used in approximately 42% of documented cases to aid cytological or histological diagnosis of common secondary thyroid cancers [2]. A combination of markers for confirming thyroid origin as well as the primary origin of the neoplasm is often recommended. It is worth noting that carcinomas of the kidney, lung, breast, and colorectum are often metastases to the thyroid gland, and their differential diagnoses using immunohistochemistry is important and listed in Table 4. Melanoma and neuroendocrine tumours are important mimics worth discussing.

Thyroglobulin and thyroid transcription factor 1 (TTF-1) are expressed in normal thyroid tissue, thyroid adenomas, and differentiated thyroid carcinomas. Thyroglobulin is the most specific and generally sensitive for well-differentiated follicular-derived thyroid carcinoma; however, some primary thyroid neoplasms have weak or absent expression of the marker. Between 70% and 80% of anaplastic thyroid carcinomas do not stain for thyroglobulin [12]. TTF-1 is a more sensitive marker and shows specific nuclear staining. However, it is positive in lung neoplasms, some central nervous system neoplasms, and small cell carcinoma and may show focal staining in some other cancers. Nevertheless, TTF-1 can often assist in differentiating primary thyroid carcinoma from secondary thyroid metastases (sensitivity 93%, specificity 96%) [78].

*Paired-box gene 8* (*PAX8*) plays an important role in thyroid, kidney, and Mullerian tract development [79]. PAX8 protein is positive in thyroid carcinoma, renal cell carcinoma, Mullerian tumours (female genital tract), and thymic tumours. As a nuclear stain, it is sensitive and expressed in 76% of anaplastic thyroid carcinomas and 100% of its squamous variants [80,81]. It is a sensitive marker for aggressive types of thyroid carcinomas, such as anaplastic follicular cell-derived thyroid carcinoma, which could be positive even when TTF-1 or thyroglobulin are negative [81,82].

PAX8 cannot be utilised to distinguish between primary cancer from renal cell carcinoma (92% positive) and primary thyroid tumours such as papillary thyroid carcinoma, which stains for PAX8 100% of the time [3,9,78]. Immunohistochemistry plays an important role in diagnosing metastatic renal cell carcinoma due to its overlapping cytomorphologic features with papillary and follicular thyroid neoplasm [3]. The immunochemical profiles of TTF-1(−), thyroglobulin (−), and carbonic anhydrase IX (CAIX) (+) have been shown to be 100% sensitive and specific in identifying clear cell renal cell carcinoma metastasis to the thyroid [83]. In addition, PAX8 positivity may also be useful in ruling out metastatic squamous cell carcinoma as it is often negative in head and neck squamous cell carcinomas and two-thirds of pulmonary squamous cell carcinomas [80].

TTF-1 is positive in both lung and thyroid carcinomas. Thus, the combination of Napsin A (positive in lung carcinoma) and monoclonal PAX8 (positive in thyroid carcinoma) has been suggested as the best panel of markers to distinguish between primary thyroid carcinomas and metastatic lung carcinomas to thyroid [12,78].

Breast carcinoma and colorectal carcinoma are also common amongst secondary cancers to the thyroid. Breast carcinoma can be identified by positivity to GATA-3 and sometimes hormone receptors, whereas colorectal carcinoma is often positive to SATB2 [84,85]. GATA-3 is a transcription factor important in the differentiation of breast epithelia, urothelial, and subsets of lymphocytes and hence can be utilised in patients with a known history of breast cancer. S-100, melan A, and HMB-45, Sox-10 are markers used for a suspected melanoma primary [3].

Metastatic neuroendocrine neoplasms of the thyroid gland should be differentiated from primary medullary thyroid carcinoma and, rarely, primary paraganglioma of the thyroid. All these tumours appear similar, with tumour cells having granular cytoplasm, vascular stroma, and positive reaction to neuroendocrine markers [47]. For differential diagnosis, medullary thyroid carcinoma is positive for calcitonin, whereas metastatic neuroendocrine thyroid carcinoma is negative. Paraganglioma is negative for cytokeratin, whereas metastatic neuroendocrine neoplasms are positive for cytokeratin [86]. In addition, paraganglioma could have S-100 positive sustentacular cells and a characteristic alveolar histological pattern [87]. Stains for tyrosine hydroxylase and GATA-3 may be positive [87].

Hence, investigating previous cancer diagnoses is critical in selecting appropriate immunohistochemistry panels. A detailed panel would be indicated in the presence of unusual histology and limited clinical history. Immunohistochemistry alone does not eliminate the need for correlation with clinical and radiographic features.

## 9. Molecular Oncology

Molecular and genetic testing is a rapidly evolving field that provides opportunities to overcome diagnostic dilemmas and for tailored antitumor therapy. The addition of next-generation sequencing, as well as the common use of fluorescence in-situ hybridisation (FISH), can facilitate the identification of molecular alterations, which could confirm the tissue origin of a metastatic deposit and predict treatment responses [88]. Table 5 summarises the molecular markers reported in cases of secondary thyroid cancer and individualised treatments in certain cases.

Knowledge of the molecular markers specific to primary thyroid cancers is useful in distinguishing such neoplasms from extra-thyroidal metastases. Papillary thyroid carcinoma could harbour *BRAF* mutations, *RET*/*PTC* rearrangements, and *RAS* mutations [57]. Mutation of the *RET* oncogene is associated with familial medullary thyroid carcinoma (including in multiple endocrine neoplasia type 2A and 2B). Thyroid follicular cell-derived carcinomas have *RAS* and *PI3K*-*AKT* pathway involvement [89]. The *PAX8*/*PPARG* translocation results in a fusion protein that inhibits the tumour suppressor effects of *PPARG*. This translocation is seen in 25% to 60% of follicular thyroid cancers and a third of follicular variant papillary thyroid carcinoma (mostly labelled now as non-invasive follicular thyroid neoplasm with papillary-like nuclear features [NIFTP]) [89].

RAS proteins are involved in cell growth and differentiation; hence, genetic mutations can lead to cancer. The three *RAS* isoforms, *HRAS*, *KRAS*, and *NRAS*, are the most common oncogenes in humans. *KRAS* mutations have been identified in secondary thyroid cancers with primary lung adenocarcinoma [12,88]. It has also been reported in several cases of colon adenocarcinoma to the thyroid [90,91,92,93]. In patients with colorectal carcinoma metastasis, *KRAS* and *NRAS* mutations are tested prior to anti-epidermal growth receptor (EGFR) target therapy, as mutated RAS proteins downstream of EGFR will impair treatment effectiveness. *KRAS* mutation is more common than *NRAS* mutation in colorectal carcinoma. Previously, only *KRAS* would be tested. However, with the advancement and ease of testing genomic mutations, both mutations in colorectal carcinoma can be tested before commencing treatment. Table 5 illustrates that testing the *RAS* mutations in metastatic colon cancer of the thyroid can help select therapy based on the mutation status. If wild type *RAS* is found, the patient could be treated with anti-EGFR antibody. On the other hand, if a mutation is detected in either *KRAS* or *NRAS*, other therapies are considered, such as conventional chemotherapy (i.e., irinotecan, capecitabine) or anti-angiogenic therapy (i.e., bevacizumab).

Epidermal growth factor receptor (EGFR) is a transmembrane glycoprotein containing tyrosine kinase activity. Mutations of *EGFR* leading to its overexpression have been associated with lung adenocarcinoma [57]. More than 60% of non-small cell lung carcinomas, predominately adenocarcinoma, show *EGFR* mutations. The mutational *EGFR* status of a patient facilitated a diagnosis of lung adenocarcinoma metastasis, resulting in targeted treatment with erlotinib or gefitinib (EGFR inhibitor) and consequently disease remission [94]. New *EGFR* mutations may lead to resistance of EGFR inhibitors. Thus, the oncologist may need to re-assess the mutation in recurrent or drug-resistant metastatic lung carcinoma from biopsies of metastatic sites.

*Anaplastic lymphoma kinase* (*ALK*) rearrangement has also been reported in non-small cell lung carcinomas, such as *Echinoderm microtubule-associated protein-like 4-anaplastic lymphoma kinase* (*EML4*-*ALK*) fusion identified in a lung adenocarcinoma metastasis to the thyroid [12]. *EML4*-*ALK* fusion is most often detected in a subset of non-small cell lung carcinoma in never smokers and has unique pathologic features. ALK inhibitors could be used as a target therapy for metastatic lung carcinoma. Table 5 shows that both *EGFR* mutations and *EML4*-*ALK* fusion have been used to test on the metastatic lung adenocarcinoma of the thyroid to decide the practical approach for therapy in cancer patients.

BRAF V600E protein expression results from a *BRAF* point mutation (T1799A) in exon 15. This leads to serine/threonine kinase activation [95]. The presence of *BRAF* V600E mutation can assist in identifying primary papillary thyroid cancer [96], but it is also common in melanoma [97]. Around 45% of papillary thyroid carcinomas and 15% of follicular variant papillary thyroid carcinomas harbour this mutation [89]. With the re-classification of papillary thyroid carcinoma based on the cancer genome atlas (TCGA), more than half of papillary thyroid carcinomas harbour the mutation. It is of particular importance as the mutation could be identified by detecting the mutated protein by immunohistochemistry. This has the advantage of lower cost in detection compared to DNA sequencing.

As *BRAF* mutation is also common in melanoma, BRAF inhibitors can shrink or slow the growth of metastatic melanoma in patients harbouring this mutation [98]. Metastatic melanoma can occur in the thyroid. Collins and colleagues reported a *BRAF* wild type mutation in thyroid cancer secondary to ocular melanoma [99].

Renal cell carcinoma is the most frequent primary origin of secondary thyroid in a surgical series [2]. Von Hippel Lindau disease, an autosomal dominant condition caused by a mutation in the *VHL* tumour-suppressor gene, is the most common type of hereditary renal cell carcinoma. In addition, the mutation is an integral component in the development of the great majority of sporadic renal cell carcinomas. In a meta-analysis of clinical trials, the *VHL* gene mutation had no prognostic or predictive value to patients who could receive anti-vascular endothelial growth factor (VEGF) therapy [100]. Nevertheless, VHL mutation was detected in two cases of thyroid cancers secondary to clear cell renal cell carcinoma metastasis [12,101].

Breast carcinoma is classified into molecular subtypes for adjuvant chemotherapy [102]. Human epidermal growth factor receptor 2 (HER-2) is a member of the EGFR family. Amplification of *HER-2* with overexpression of this oncogene is a molecular subtype of the breast, which has aggressive clinical behaviour and susceptibility to HER-2 kinase inhibitor therapy. Liu and colleagues described a patient with infiltrative ductal carcinoma of the breast metastatic to the thyroid gland which showed positive oestrogen receptor expression and *HER-2* gene and protein expression. Hence, this patient underwent a clinical trial using afatinib (EGFR and HER-2 kinase inhibitor) [103].

**Table 5 ijms-23-03242-t005:** Molecular markers of secondary thyroid cancer cases published in literature.

Author (Year)	Primary Cancer	Molecular Markers	Systemic Therapy
Ghossein et al. (2020) [12]	Lung adenocarcinoma (*n* = 3)	*EML4-ALK* fusion (*n* = 1), *KRAS* mutation (*n* = 2) in thyroid metastasis	
Ko and Kim (2020) [104]	Lung adenocarcinoma	*EGFR* (exon21: L858R) mutation in primary site and thyroid metastasis. No mutation in *ALK*, *PD-L1*, *BRAF*.	Gefitinib
Yamada et al. (2020) [105]	Lung adenocarcinoma	*EGFR* G719X mutation in primary site and thyroid metastasis	
Bellevicine et al. (2015) [88]	Lung adenocarcinoma	*KRAS* G12C mutation in primary site and thyroid metastasis	Chemotherapy
Albany et al. (2011) [94]	Lung adenocarcinoma	*EGFR* L858R mutation in primary site	Erlotinib
Hashimoto et al. (2011) [70]	Lung adenocarcinom ^a^	Primary site and lung component in thyroid: *EGFR* (exon21: L858R) mutation FVPTC component in thyroid: no *EGFR* mutation No *BRAF*, *NRAS* and *HRAS* mutation in any site	Gefitinib
Mitani et al. (2020) [106]	Suspected non-small cell lung carcinoma	*EGFR* mutation (L858R and T790M) in thyroid metastasis No *BRAF(V600E)* mutations	Erlotinib then Osimertinib
Liu et al. (2014) [103]	Infiltrating duct carcinoma (no special type) of breast	*HER-2* amplification in primary site and thyroid metastasis	Afatinib trial
Ghossein et al. (2020) [12]	Clear cell renal cell carcinoma	*VHL* mutation in thyroid metastasis	
Bugalho et al. (2006) [101]	Clear cell renal cell carcinoma	*VHL* 680 delA (codon 156/exon 3) mutation in primary site and thyroid metastasis	
Yu et al. (2009) [68]	Clear cell renal cell carcinoma ^b^	RCC component in thyroid: loss of heterozygosity at *VHL* locus (3p.25–26) and chromosomal regions 3p, 5q, 14q, and 17q FVPTC component in thyroid: *NRAS* 61 mutation No *BRAF* mutation or *RET/PTC* rearrangement in any site	Chemotherapy
Keranmu et al. (2017) [90]	Colon adenocarcinoma	*KRAS* wild type in thyroid metastasis	Monoclonal anti-EGFR antibody and irinotecan
Lecumberri et al. (2017) [91]	Colon adenocarcinoma	*KRAS* mutation in thyroid metastasis	Chemotherapy
Payandeh et al. (2016) [92]	Colon adenocarcinoma	*NRAS* wild type, KRAS codon12-mutation p Gly12Asp (c.35G>A) in thyroid metastasis	Bevacizumab and capecitabine
Cozzolino et al. (2009) [93]	Colon adenocarcinoma	*KRAS* G12D mutation in primary site and thyroid metastasis	
Collins et al. (2016) [99]	Ocular melanoma	Chromosome 3 monosomy in primary site, *BRAF* wild type in thyroid metastasis	Pembrolizumab (immunotherapy)
Xing et al. (2021) [107]	Myxoid liposarcoma of thigh	*TERT* promotor and *PIK3CA* mutations in thyroid metastasis	
Ghossein et al. (2020) [12]	Malignant giant cell tumour of femur	*H3F3A G34W* mutation in thyroid metastasis	
Murro et al. (2015) [108]	Synovial sarcoma of trachea-oesophageal groove *	Translocation (X;18) (p11; q11) in thyroid metastasis	Carboplatin and paclitaxel
Afrogheh et al. (2016) [66]	Endometrial adenocarcinoma ^c^	Primary site and endometrial component in thyroid: *PTEN* (c.494G>A), *PIK3CA* (c.3132T>A), *CTNB1* (c.1661G>C) mutation and chromosome 1q amplification Hürthle cell adenoma component in thyroid: molecular markers not specified	

FVPTC: follicular variant of papillary thyroid carcinoma. ^a^ Collision tumour of metastatic lung adenocarcinoma and FVPTC. ^b^ Collision tumour of metastatic renal cell carcinoma and FVPTC. ^c^ Collision tumour of metastatic endometrial adenocarcinoma and Hürthle cell adenoma in the thyroid. * Direct extension into thyroid gland, not distant metastasis.

Other than common metastatic carcinoma and melanoma, molecular alterations have been noted in metastatic sarcoma to the thyroid gland. In giant cell tumours of bone, up to 95% harbour the *H3F3A G34W* gene mutation [109]. The identification of this mutation was diagnostic for metastasis of a giant cell tumour of the femur to the thyroid [12]. Metastasis of myxoid liposarcoma of the thigh to the thyroid was found to have *TERT* promotor and *PIK3CA* mutations [107]. *TERT* promoter mutations reduce telomerase activity, while *PIK3CA* mutations stimulate cell growth. *PIK3CA* encodes p110 alpha protein, a subunit of the PI3K enzyme, which is used in cell signalling for proliferation, migration, and survival [110]. Murro and colleagues describe a direct extension of synovial sarcoma of the tracheoesophageal groove into the thyroid gland with translocation (X;18) (p11; q11) identified on FISH [108]. Approximately 90% of synovial sarcomas harbour this translocation resulting in the *SYT*-*SSX* gene fusion product, which is rarely found in other tumours [111].

Molecular techniques can help diagnose tumour-to-tumour metastases. Yu and colleagues describe a case of metastatic renal cell carcinoma to follicular variant papillary thyroid carcinoma where molecular testing was required as primary site material was unavailable and immunohistochemical stains did not provide a definitive diagnosis [68]. Within the poorly differentiated component of the collision tumour, loss of heterozygosity in a *VHL* gene locus, and various chromosomal regions were identified. Conversely, these alterations were not observed in the follicular variant papillary carcinoma component of the tumour and, instead, an *NRAS* 61 mutation was detected. These findings distinguished the two components as having different clonal origins. In other case reports, molecular testing was also shown to assist in diagnostic confirmation after histopathological and immunohistochemical analysis [66,70].

MicroRNAs regulate around one-third of the human genome and may operate as oncogenes or tumour suppressor genes [89]. Studies have shown promising results of microRNA in predicting the origin of unknown primary cancers; however, there are no secondary thyroid cancer cases in the literature currently describing the use of this [112,113].

Molecular and genetic profiling appear to be underutilised in the diagnosis of secondary thyroid cancers due to limited cases and costs involved; hence, it is difficult to formally evaluate the value of such investigations. Current literature lacks a complete review of ancillary biomarkers for secondary thyroid cancer [12]. Further research is recommended to examine the molecular characteristics of primary cancers and metastatic sites and guide treatment options [114]. As illustrated in Table 5, molecular profiling of neoplasms will allow clinicians to better select patients who are suitable for targeted therapies [93]. Systemic therapies will be further discussed in this paper (see Section 11).

## 10. Prognosis

The prognosis and management of secondary thyroid cancer depend on the primary site, extent of metastases, and clinical symptoms. Advanced stage and multiple metastases are often associated with poor prognosis [2,7]. A large proportion of patients with metastatic cancer to the thyroid have widely disseminated disease on diagnosis. One report demonstrated the presence of widely metastatic disease in 100% (*n* = 13) of patients at the time of diagnosing secondary thyroid cancer [13]. A large study by Romero Arenas and colleagues examined 90 patients and found that 12% had isolated metastatic disease to the thyroid, 51% had involvement of nearby cervical nodes and/or tissues, 18% had other distant metastasis other than the thyroid, and 19% had an unresectable primary tumour and thyroid gland [6]. On average, the thyroid is the sole metastatic site in 17% of secondary thyroid cancer cases from recent clinical series, and up to 50% have been reported at a single centre [6,7,12,21,22]. Those with isolated metastasis to the thyroid have less disease burden and are more likely to be offered surgical resection rather than systemic therapy or palliative care. Hence, the varying stages of cancer at presentation means that a thorough workup is required to ensure tailored treatment to individuals.

The stage, aggressiveness, and associated complications of the primary cancer are the key factors affecting survival in those with metastatic cancer to the thyroid. Renal cell carcinomas are known to have a more indolent course of disease compared to other cancers such as lung cancer. In one study amongst 36 cases of secondary thyroid cancer, researchers reported the shortest survival time for an oesophagus primary (2 months) while the longest survival was in a patient with renal cell carcinoma (20 months) [7]. Calzolari and colleagues analysed 25 surgical cases of secondary thyroid cancer and reported the 5-year survival rate of those with renal cell carcinoma primaries to be 40% compared to 0% for other primary cancers, including those from the lung, colon, breast, melanoma, and unknown origin [15]. Out of 16 patients with primary cancer of the lung metastasising to the thyroid, nearly all died within 24 months of diagnosis, and only three were offered thyroidectomy [6]. Patients with cancer of unknown origin often have poor prognoses with a median survival of less than 12 months.

Roughly up to half of all deaths are caused by the malignancy itself, such as asphyxiation from a head and neck primary tumour [18,19]. One systematic review reported 31% of non-surgical secondary thyroid cancer patients died of complications related to the cancer [2]. The other half of all deaths are attributable to cardiovascular disease (myocardial ischemia/infarction, coronary thrombosis), pulmonary disease (pulmonary embolism, pneumonia, acute respiratory distress syndrome), and others (peritonitis, cerebral stroke) [18].

## 11. Treatment

### 11.1. Surgical Intervention

Surgical resection has been shown in some studies to achieve long term survival in patients with a treatable primary tumour and isolated metastasis. It may also be combined with radiotherapy, chemotherapy, or hormone therapy to reduce the risk of recurrence. A meta-analysis by Straccia and colleagues demonstrated that amongst patients with disseminated disease, total thyroidectomy increased disease-free interval and overall survival compared to chemotherapy and/or radiotherapy alone. In a systematic review, the median-free survival period for surgical patients was 65 months compared to 13 months for non-surgical patients who received either chemotherapy, radiotherapy, or no treatment [2]. It is important to note that 44% of the surgical patient group also received adjuvant chemotherapy or chemotherapy combined with radiotherapy. Similarly, a retrospective study of 90 cases found that the median overall survival was longer in patients who underwent thyroidectomy compared to no surgical intervention (34 months vs. 11 months; *p* < 0.0001) [6].

Conversely, other studies have reported that although surgery lengthened time from diagnosis to death, it did not affect overall survival [7]. Furthermore, there are discrepancies in survival statistics between different primary cancer types. Patients with renal cell carcinoma primary cancer often have better outcomes compared to melanoma, lung, and oesophageal primary cancers, which have a more aggressive clinical course [6,37]. The 5-year survival rate for those with renal cell carcinoma primaries was 40%, whereas no patients with lung, colon, breast, and melanoma primary cancers survived to five years [15]. Hence, promising results regarding survival in surgical patients may only be applicable to those with relatively indolent tumours, such as renal cell carcinoma. The interpretation of results is further limited as surgical candidates are selected based on their stage of disease and comorbidities. Surgery is not warranted in patients who are unfit for surgery or have widespread disease. Healthier patients with isolated metastatic disease tend to be ideal surgical candidates. Prospective trials are lacking, and amongst the limited number of retrospective studies available, there is significant heterogeneity in patient populations and study designs.

Morbidity from surgical resections has been reported in few studies. Between 0% and 16% of cases experienced surgical-related complications [11,15]. Out of 25 cases, three resulted in hypoparathyroidism (one permanent, two transient), and one case had permanent recurrent nerve palsy [15]. A literature review reported one case of transient stridor and three cases of hypocalcaemia post-resection [115]. Other risks related to surgery include aspiration pneumonia or injury of surrounding structures, which may require the insertion of a tracheostomy or percutaneous enteral gastrostomy [115]. There have been no reported mortalities associated with surgery.

Regarding lymph node dissection, multiple factors are taken into consideration when deciding on the appropriateness of the procedure. Literature on the role of lymph node dissection in patients with metastasis to the thyroid is lacking.

### 11.2. Systemic Therapy

In those with disseminated disease, systemic therapy may be used alone or as adjuvant therapy [96]. The decision to commence a patient on chemotherapy as sole or adjuvant treatment depends on various patient and clinician factors. Furthermore, target therapies are becoming more prevalent due to their specificity to certain cancers and improved side effect profiles compared to chemotherapy. Hence, literature is limited regarding the use of chemotherapy in secondary thyroid cancer. In general, patients with primary cancers responsive to chemotherapy may be considered as candidates for treatment upon weighing the risks, benefits, and alternatives, such as surgery or other systemic treatments. Chemotherapy is generally used for those with widespread distant metastatic disease, which cannot be treated with surgical resection alone. Historically, cancer of unknown origin is also treated with empirical chemotherapy; however, such patients generally have poor prognoses. Certain primary cancers do not respond well to chemotherapy. Metastatic renal cell carcinoma, the most common cancer metastasising to the thyroid gland, is usually resistant to chemotherapy and radiotherapy. Prior to tyrosine kinase inhibitors, these patients were treated with cytokine interferon alpha or interleukin-2 (IL-2).

Lung and breast carcinomas are also common cancers that metastasise to the thyroid gland. In recent years, treatment for non-small cell lung carcinoma has evolved from chemotherapy to targeted therapy based on genetic profiling. Chemotherapy may also be used to treat metastasis from the breast, whereas endocrine therapy and targeted therapy can be used across disease stages. Hormonal and targeted therapy can offer tailored treatment to the individual and are generally better tolerated by patients compared to standard chemotherapy.

Hormonal therapy may be reserved for breast or prostate primary cancers that are responsive to treatment. One case of metastatic lobular type breast carcinoma to the thyroid initially underwent a total thyroidectomy [116]. Immunohistochemical staining was positive for oestrogen receptor, progesterone receptor, and GATA3. Given these findings, the patient was further treated with an aromatase inhibitor. In addition, Wang and colleagues described a metastatic invasive ductal breast carcinoma to the thyroid, which was positive for HER-2 and GATA 3 and negative for oestrogen and progesterone receptor. The patient received paclitaxel liposome and trastuzumab (monoclonal antibody specific to HER-2) and was alive on follow up at 19 months post-diagnosis of metastasis [117].

Targeted therapies with small molecule inhibitors or immunotherapy have resulted in remission in some cases. Tyrosine kinase activity is present in receptors such as epidermal growth factor receptor (EGFR) and vascular endothelial growth factor receptor (VEGFR). Patients with mutations are more likely to respond to receptor-specific inhibitors. In lung cancer, *EGFR* is a common targetable mutation. Albany and colleagues reported a case of metastatic lung adenocarcinoma to the thyroid, which was treated with erlotinib (epidermal growth factor receptor tyrosine kinase inhibitor) upon discovery of an L858R *EGFR* mutation in the primary. Clinical improvement was seen at four weeks, and remission was noted at the eight-month follow up [94]. Similarly, a case of secondary thyroid cancer with unknown primary initially underwent empirical chemotherapy (paclitaxel and carboplatin), which yielded no response. A total thyroidectomy was performed, and genetic profiling of the specimen revealed L858 and T790M *EGFR* mutations. Therapy was adjusted from erlotinib to osimertinib (another epidermal growth factor receptor tyrosine kinase for specific *EGFR* mutation) based on this, which resulted in clinical improvement [106]. In another case report, a patient with metastatic uveal melanoma to the thyroid was found to have *BRAF* mutation wild type [99]. The patient was commenced on ipilimumab, which had no effect on disease progression and hence was switched to pembrolizumab (a novel immunomodulator) a year later, resulting in disease stabilisation.

Colorectal adenocarcinoma is another common primary tumour metastasised to the thyroid gland. A metastatic colorectal adenocarcinoma reported in Iran was found to have *NRAS* wild type status but a *KRAS* codon12-mutation (p Gly12Asp (c.35G>A)), deeming the patient ineligible for cetuximab (EGFR inhibitor) [92]. Hence, the patient received combined capecitabine and bevacizumab (VEGF inhibitor) and experienced improvement in symptoms after 6 months. The small number of case reports in the literature highlight the utility of genetic profiling in selecting patients for targeted therapies such as tyrosine kinase inhibitors, which may be more effective and specific in the treatment of metastatic cancer.

Targeted therapy, however, also depends on the primary cancer type and extent of disease. A patient post-chemotherapy for metastatic colorectal adenocarcinoma to the liver was found to have additional metastasis to the thyroid gland on follow up [90]. As the thyroid tumours showed wild-type *KRAS*, the patient was trialled in a study on recombinant chimeric monoclonal anti-EGFR antibody and irinotecan. Unfortunately, the disease progressed, and the patient died within 5 months due to organ failure.

With the increasing advancement of tyrosine kinase inhibitors and immune checkpoint inhibitors, surgery could eventually be reserved for palliative treatment in certain cases. One case report describes a patient with disseminated metastatic renal cell carcinoma who underwent a hemithyroidectomy followed by a course of sunitinib [118]. This patient showed a complete response to treatment at five months. The CARMENA trial demonstrated that treatment with sunitinib alone in patients with metastatic renal cell carcinoma did not result in inferior survival compared to those receiving cytoreductive nephrectomy followed by sunitinib [119]. This indicates that those with metastatic renal cell carcinoma requiring systemic therapy do not benefit from cytoreductive nephrectomy and, in fact, also experience harm from surgical intervention. Interestingly, the more recent CLEAR trial found that lenvatinib combined with pembrolizumab resulted in significantly longer progression-free survival and overall survival in patients with metastatic renal cell carcinoma compared to sunitinib only [120]. Hence, obtaining a histological diagnosis is now imperative as targeted treatments may be better tolerated by patients than surgery or standard chemotherapy. Within this era of targeted therapy, further clinical trials comparing different treatments will further guide management and tailoring to individuals.

### 11.3. Other Interventions and Symptomatic Measures

Radiotherapy appears to be of limited usage in the treatment of disseminated disease; however, it can be useful in localised or residual disease. One case report described the use of stereotactic radiotherapy in a patient post-total thyroidectomy for metastatic adenoid cystic carcinoma of the lung to the thyroid gland with nerve and trachea infiltration [121]. There were no signs of oncological relapse on repeat imaging within one year. However, a retrospective review by Kim and colleagues concluded that although the use of radiotherapy provided symptom relief, it did not prevent the progression of disease [16].

Surgical interventions can also be used for local disease control and symptom relief. For example, tracheostomy insertion or debulking surgery can relieve airway compression [17]. Ishikawa and colleagues reported poor survival in 3 of 4 cases after thyroidectomy; however, surgical intervention improved quality of life by preserving swallowing and breathing functions [8]. Chemotherapy and radiotherapy were reported to not provide any benefit in local disease control amongst palliative patients [7].

### 11.4. Management in Recent Clinical Series

The management of secondary thyroid cancer varies based on the primary cancer and individual patient. Figure 2 provides an overview of the assessment and management of potential metastasis to the thyroid. Many receive multimodal treatment for the primary cancer and its metastases. This includes surgery, radiotherapy, chemotherapy, and/or targeted therapy. A large proportion of patients undergo surgical resection for the primary tumour. For instance, those patients with a primary from renal cell carcinoma (89%) were more likely to have surgical resection of the primary than those with lung (25%) or head and neck (47%) cancer [6]. In addition, amongst 147 cases of metastatic clear cell renal cell carcinoma to the thyroid, 80.3% were treated with surgical resection, 12.2% with systemic therapy (including tyrosine kinase inhibitor, interferon alpha, IL-2, non-specific chemotherapy), 2.7% received radiotherapy, and 1.4% did not undergo any treatment [17]. Conversely, one study involving eight cases of metastatic lung adenocarcinoma to the thyroid demonstrated that 50% received chemotherapy, 12.5% received both chemotherapy and radiotherapy, 12.5% underwent a total thyroidectomy, and 25% received no treatment [122]. Amongst 21 cases of secondary thyroid cancer reported in China, 14 were treated with chemotherapy and/or radiotherapy, and only 1 received a partial thyroidectomy [21]. Most of these cases originated from the oesophagus, followed by the breast, head and neck, and unknown site. These examples illustrate the factors that clinicians consider when deciding patient suitability for surgery, including cancer aggressiveness, prognosis, and likely clinical outcomes based on surgical or systemic interventions.

## 12. Conclusions

Secondary cancer to the thyroid gland is a rare condition that may be discovered incidentally and poses several diagnostic challenges. FNA or surgical resection with histological and immunohistochemical analysis are the main methods used to confirm diagnosis. Establishing criteria to identify cases suspicious for secondary thyroid cancer is recommended (i.e., clinical or radiological evidence of a thyroid mass, history of non-thyroidal malignancy). Biopsies and ancillary testing are means for ongoing research regarding the molecular parameters of various primary tumours.

## Figures and Tables

**Figure 1 ijms-23-03242-f001:**
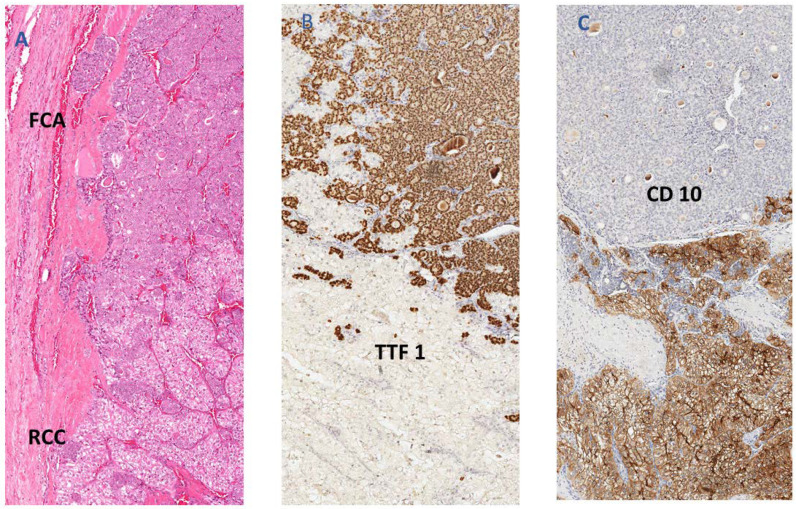
(**A**) Histology of a collision tumour in thyroid with primary follicular carcinoma (FCA) and metastatic renal cell carcinoma (RCC); (**B**) TTF-1 immunohistochemical stain is positive in the follicular cell carcinoma component; (**C**) CD10 immunohistochemical stain is positive in the renal cell carcinoma component.

**Figure 2 ijms-23-03242-f002:**
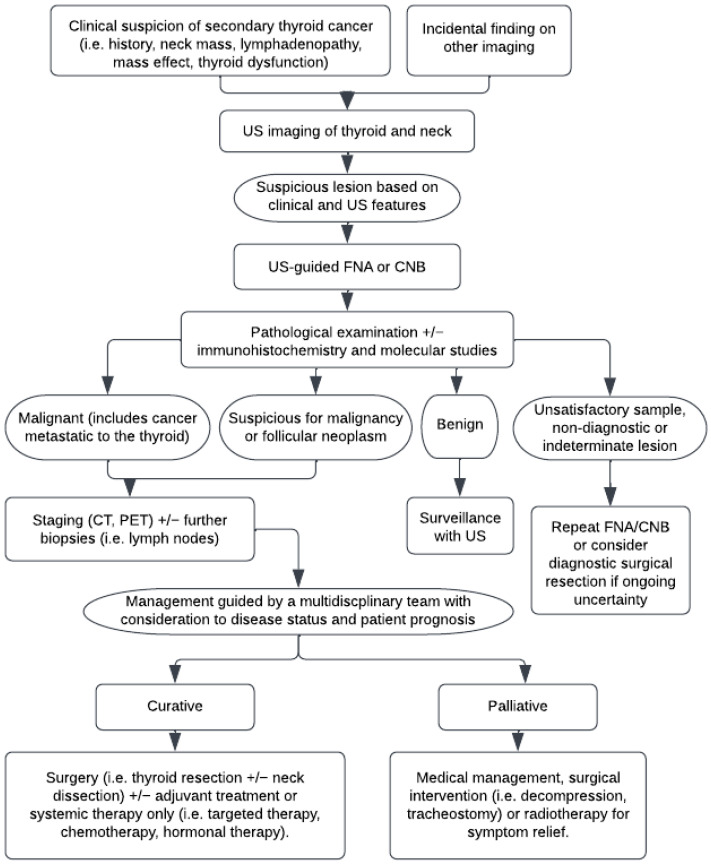
An overview of the assessment and management of potential metastasis to the thyroid. US, ultrasound; FNA, fine-needle aspiration; CNB, core-needle biopsy; CT, computed tomography; PET, fluorodeoxyglucose-positron-emission tomography.

**Table 2 ijms-23-03242-t002:** Cytological diagnoses of secondary thyroid cancers by fine needle aspiration.

Author (Year)	No. of FNAs Performed on STCs	Proportion of STCs Diagnosed Correctly as Metastasis on FNA	Proportion of STCs Diagnosed as Suspicious for Malignancy/Unknown Primary on FNA	Proportion of STCs with Other Diagnoses on FNA
HooKim et al. (2015)	28	85.7%	14.3%	0%
Hegerova et al. (2015)	97	94%	0%	6% follicular neoplasm or papillary thyroid carcinoma
Pusztaszeri et al. 2015	62	81%	0%	8% nondiagnostic 5% follicular neoplasm 7% primary thyroid cancer
Choi et al. (2016)	41	46.3%	24.4%	4.9% benign 9.8% nondiagnostic 4.9% atypia or follicular lesion of undetermined significance 9.8% others
Straccia et al. (2017)–meta-analysis	154	72.7%	26.6%	0.65% primary thyroid cancer
Zivaljevic et al. (2018)	6	33%	Nil data provided	Nil data provided
Stergianos et al. (2021)	29	41.4%	Nil data provided	0.07% follicular neoplasm 0.03% benign 0.03% papillary thyroid cancer 0.03% anaplastic thyroid cancer

FNA, fine-needle aspiration; STC, secondary thyroid cancer.

**Table 4 ijms-23-03242-t004:** Immunohistochemical profiles for common differential diagnoses of metastatic cancer in the thyroid versus primary thyroid neoplasms.

Metastatic in Thyroid with Primary Tumour From	Main Differential Diagnoses	Usual Immunochemical Profiles	
		Positive in Metastatic Primary Tumour	Positive in Thyroid, Primary Thyroid Neoplasms, or Parathyroid
Renal cell carcinoma	Oncocytic (Hürthle) cell neoplasm, follicular neoplasm with clear cell component	CAIX, CD10	TTF-1, thyroglobulin
Lung carcinoma (adenocarcinoma)	Anaplastic follicular cell-derived thyroid carcinoma	Napsin	PAX-8
Breast carcinoma	Follicular thyroid neoplasms, parathyroid neoplasms	GATA-3, oestrogen receptor, progesterone receptor	TTF-1, thyroglobulin
	Parathyroid neoplasms	GATA-3, oestrogen receptor, progesterone receptor	Parathyroid hormone
Colorectal adenocarcinoma	Anaplastic follicular cell-derived thyroid carcinoma	SATB2	TTF-1, thyroglobulin
Malignant melanoma	Papillary thyroid carcinoma, anaplastic follicular cell-derived thyroid carcinoma	S-100, HMB-45, Sox-10	TTF-1, thyroglobulin,
	Medullary thyroid carcinoma	S-100, HMB-45, Sox-10	Calcitonin
Neuroendocrine tumour	Medullary thyroid carcinoma	Cytokeratins	Calcitonin
	Paraganglioma in thyroid	Cytokeratins	S-100 (sustentacular cells), tyrosine hydroxylase, GATA3
Squamous cell carcinoma of the head and neck	Anaplastic follicular cell-derived thyroid carcinoma, squamous subtype	-	PAX-8, TTF-1, thyroglobulin (only in a portion of cases)

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
