# Peer review of "Clinicopathological and Molecular Features of Secondary Cancer (Metastasis) to the Thyroid and Advances in Management"

_ijms, 2022, doi:10.3390/ijms23063242_

Round 1

Reviewer 1 Report

Marie Nguyen et al. provide a comprehensive review of the current knowledge on the treatment of secondary thyroid cancer. The work is interesting and well written. I only have some minor comments:

Line 10 -  Collison – did you mean collision or collison?

Line 259 - the original primary site – some minor text editing is needed

Line 304 – ROSE – please add one or two more sentences describing the procedure. Is this an intra-operative procedure or can it be used during standard biopsy?

Line 337 – recipient/ benign recipients – please explain the term “recipients”

Line 374 – primaries – consider using terms like “primary cancers” instead of “primaries”

Author Response

Marie Nguyen et al. provide a comprehensive review of the current knowledge on the treatment of secondary thyroid cancer. The work is interesting and well written. I only have some minor comments:

Line 10 - Collison – did you mean collision or collison?

Response: Change to Collision

Line 259 - the original primary site – some minor text editing is needed

Response: Change as suggested

Line 304 – ROSE – please add one or two more sentences describing the procedure. Is this an intra-operative procedure or can it be used during standard biopsy?

Response: ROSE is a procedure to provide a pathology staff at the site of fine needle aspiration which allow the assess the adequacy of the material of cytology smears to increase the yield of the material.  The descriptions are added as suggested 

Line 337 – recipient/ benign recipients – please explain the term “recipients”

Response: The descriptions of collision tumour regarding definition of donor and recipient was added in Session 6.

Line 374 – primaries – consider using terms like “primary cancers” instead of “primaries”

Response: Change as suggested

Reviewer 2 Report

Well-written review of diagnostic and clinical work-up of metastases to the thyroid. Clinically relevant topic, well summarized in comprehensive tables. I have some suggestions to improve the study even more:

  1. There is a recent case series lacking, and I believe it should be included as it is one of the largest to date; PMID: 33642087. Please also update Table 1 with this reference in mind. I cannot see why the review period should end abruptly at 2020?
  2. There are no figures in this submission, which is a draw-back. I suggest the authors include a metastatic case and present the main findings from for example US investigations, gross and microscopic pathology including IHC, to illustrate the study and make it more easy to digest the sheer text mass.
  3. In a similar fashion, the concluding remarks from the authors could also be highlighted in a figure using a very simple flow-chart in how to assess a potential thyroid metastasis from imaging, pathology, surgery/oncology perspectives.
  4. Table 4 may also include head&neck carcinomas (metastatic squamous cell carcinoma), it would be interesting to hear the authors' take on differentating met SCC from subsets of ATC in terms of morph/IHC.

Minor queries:

Text: Grave's disease --> Graves' disease.

Table 4: TT1? ---> TTF1? Also, oestrogen or estrogen?

Table 4: Headings are confusing. What is "Positive in metastatic primary tumour in thyroid"? Maybe remove "in thyroid"? Or change to "Positive in primary tumor metastatic to thyroid"? This column also suggests cytokeratins are expressed in paraganglioma, which is usually not seen.

Gene names should be in italics throughout.

Row 561: "...the most common cancer metastasise to the thyroid gland" ---> metastasizing

Author Response

Well-written review of diagnostic and clinical work-up of metastases to the thyroid. Clinically relevant topic, well summarized in comprehensive tables. I have some suggestions to improve the study even more:

  1. There is a recent case series lacking, and I believe it should be included as it is one of the largest to date; PMID: 33642087. Please also update Table 1 with this reference in mind. I cannot see why the review period should end abruptly at 2020?

Response: The series PMID: 33642087 was updated in Table 1 and the review period extend to 201

2. There are no figures in this submission, which is a draw-back. I suggest the authors include a metastatic case and present the main findings from for example US investigations, gross and microscopic pathology including IHC, to illustrate the study and make it more easy to digest the sheer text mass.

A collision tumour was added to illustrate the microscopic and immunohistochemical staining

3. In a similar fashion, the concluding remarks from the authors could also be highlighted in a figure using a very simple flow-chart in how to assess a potential thyroid metastasis from imaging, pathology, surgery/oncology perspectives.

Response: A flow chart was introduced

4. Table 4 may also include head & neck carcinomas (metastatic squamous cell carcinoma), it would be interesting to hear the authors' take on differentiating met SCC from subsets of ATC in terms of morph/IHC.

Response: metastatic SCC vs SCC subset of ATC added in Table 4 as suggested

Minor queries:

Text: Grave's disease --> Graves' disease.

Response: Changes as suggested

Table 4: TT1? ---> TTF1? Also, oestrogen or estrogen?

Response: Updated the typo in Table 4 as suggested

Table 4: Headings are confusing. What is "Positive in metastatic primary tumour in thyroid"? Maybe remove "in thyroid"? Or change to "Positive in primary tumor metastatic to thyroid"? This column also suggests cytokeratins are expressed in paraganglioma, which is usually not seen.

Responses: Changes as suggested.   This paragraph means metastatic neuroendocrine tumour is positive for cytokeratin whereas primary thyroid paraganglioma is negative but positive for S-100 and tyrosine hydroxylase, GATA3

Gene names should be in italics throughout.

Response: Gene names were checked an made italic throughout

Row 561: "...the most common cancer metastasise to the thyroid gland" ---> metastasizing

Response: Changes as suggested

Round 2

Reviewer 2 Report

Excellent revision, this is a great paper.